# Computational Simulation of Wind Microclimate in Complex Urban Models and Mitigation Using Trees

Azin Hosseinzadeh  and Amir Keshmiri *

Department of Mechanical, Aerospace and Civil Engineering, University of Manchester, Manchester M13 9PL, UK; Azin.Hosseinzadeh@manchester.ac.uk
* Correspondence: A.Keshmiri@manchester.ac.uk

**Abstract:** Due to a rapid increase in urbanisation, accurate wind microclimate assessment is of crucial importance. Evaluating wind flows around buildings is part of the planning application process in the design of new developments. In this study, computational fluid dynamics (CFD) simulations are carried out for a case study, representing the East Village in the London Olympic Park. Following a validation test against experimental data for a simpler urban configuration, the key input parameters, including appropriate boundary conditions, mesh setting and type of turbulence model, are selected for the Olympic Park model. All the simulations are conducted using the commercial code STARCCM+ under steady-state conditions with the Reynolds-averaged Navier–Stokes (RANS) method. The turbulence is modelled using different common variants of eddy-viscosity models (EVMs) including standard k-$\varepsilon$, realizable k-$\varepsilon$ and standard and shear stress transport (SST) k-$\omega$. The results demonstrate that standard and realisable k-$\varepsilon$ models correlate very well with the experimental data, while some discrepancies are found with standard and SST k-$\omega$. Following the determination of areas of high velocity, appropriate tree planting is proposed to overcome the effect of corner and downwash acceleration. With the optimised arrangement of trees and using specific types of tree (e.g., birch), wind speeds at the pedestrian level are reduced by 3.5, 25 and 66% in three main regions of interest. Moreover, we investigate the effects of tree heights. The obtained results illustrate that the wind velocity reduces when the crowns of the trees are located closer to the buildings and the ground. Our high-resolution CFD simulation and results offer a quantitative tool for wind microclimate assessment and optimised design and arrangement of trees around buildings to improve pedestrian comfort.

**Keywords:** wind microclimate; wind assessment; computational fluid dynamics; building engineering; turbulence modelling; vegetation; urban design



## 1. Introduction

A rapid increase in the construction of high-rise buildings creates the need for wind microclimate assessment. Essentially, wind microclimate assessment is performed during the design stage to show the results of wind impact on the design, which is followed by proposing methods to mitigate the wind in areas of high velocities for pedestrian comfort. Assessment of wind conditions around buildings is conducted using observational techniques and computational fluid dynamics (CFD) methods. Observational techniques are performed through measurement (e.g., wind tunnel testing) and are widely used for validation of simulations conducted using CFD techniques.

However, not all cases can be validated. Wind tunnel experiments for real cases with complex geometry are problematic due to the difficulty related to the similarity of real wind conditions with wind tunnel chambers [1]. To use this method, a scaled model of the real case needs to be built. The conditions of the wind tunnel test need to resemble the real conditions of the site in an urban area. Despite the difficulty of similarity law, when using measurement techniques all points cannot be measured in space and the results depend on

where the sensors are located. On the other hand, with numerical simulation, all the points and variables in defined computational domains can be evaluated [2]. Nowadays, with the help of computational aids, there has been growing interest in the use of CFD methods to predict the winds around buildings, and numerous works have been conducted in this area for generic or real cases [3–5].

Among them are some works that have been validated for complex geometries using wind tunnel testing, such as the work that was conducted by Taota [6]. In this work, a 3D Reynolds-averaged Navier-Stokes (RANS) model with a renormalisation group (RNG) k-ε model was used, which gave a reasonable result that fitted experimental data. The accuracy of the work was examined using the measurement data. Another study for pedestrian-level wind undertaken in Toronto by Adamek [7] evaluated pedestrian comfort and used wind tunnel testing to validate the results.

In this study, large eddy simulation (LES) was used for simulation of the flow, which is computationally more expensive than RANS methods. Another study conducted for the urban environment by Tominaga [8] included the use of five different turbulence models including standard k-ε, RNG k-ε, realizable k-ε, standard k-ω and k-ω shear stress transport (SST) under steady and unsteady state RANS methods. Although this work can be used as a good comparative study, it was only validated for a single high-rise building and it is still not clear whether or not, the difference between each method for complex geometry is negligible.

While several of the publications reviewed above have validated their results against experimental data, there are other works without validation for street canyon and generic building blocks, most of which involved RANS methods using standard and modified versions of the k-ε model, which are listed in the review paper by Blocken [3]. CFD analysis over urban areas without validation includes comparative studies where various urban configurations and design parameters are compared. Among these works, none of them compared the effects of various turbulence methods on the same configuration, same wind direction and same grid size for complex geometry. Different results are obtained using different models but the quantitative variation is not clearly mentioned in any previous works.

Therefore, the main aim of this work is to show the impact of various factors that can affect the numerical simulations around an urban environment. To do so, wind microclimate assessment for the East Village of the London Olympic Park, consisting of 67 blocks, is simulated using CFD. To validate this case, initially a CFD simulation of the wind speed between two buildings is evaluated. The results of the simulation vary depending on the input parameters includinggrid size, type of grid (e.g., structured or unstructured), the ratio of prism layer thickness to the cell size far from the walls, type of turbulence model, methods of solving the near-wall velocity (e.g., low Reynolds number or wall function), solver setting, shape and size of the computational domain. The results of the CFD simulation for two buildings are validated by the work that was done by Blocken [9]. After validation, the same settings for the solver and input parameters are used for the case of the East Village. Following the determination of areas of high velocities, planting trees as a common mitigation technique is proposed in those regions to diminish the effect of corner acceleration and downwash for the case of the East Village.

## 2. Problem Definition and Methods

### 2.1. Test Case 1: CFD Simulation of Wind Speed between Parallel Buildings

This section starts with the CFD analysis of wind speed for a generic configuration consisting of two parallel buildings. The input parameters (e.g., boundary conditions), mesh information (e.g., ratio of prism layer thickness to core cell size, number of prism layers), turbulence model and solver settings including velocity-pressure coupling are the same for this test case and for the East Village (i.e., Test Case 2, to be discussed below). The results of this test case are validated with the experimental data reported in Blocken [9]. In the present work, a cylindrical domain is used as the computational domain in contrast to

the work of Blocken where a block was used. The cylindrical domain was chosen to directly link and relate the numerical procedures in Test Case 1 to those in Test Case 2, where in the latter a circular domain was adopted to ensure adaptability to various wind directions.

Computational Domain and Mesh

The computational domain employed in this study is a cylinder with a circular sub-domain, to facilitate the application of wind direction. While in the case of using two buildings, there is only one inlet (i.e., the wind is blowing from one direction), in reality, and for a real urban geometry, the wind could be coming from different directions.

Based on guidelines for CFD simulations of pedestrian comfort [10,11], the outflow boundaries must be 15 $H_{max}$, where $H_{max}$ is the height of the tallest building. The top and inlet boundaries must be at least 5 $H_{max}$ from the target area. The real scale building has a height of 10 m. The radius and height of the chosen cylindrical computational domain here are given as 150 m and 50 m, respectively. The building blocks are located within a smaller circular subdomain to distinguish between the mesh size in the vicinity of the area of interest and far from it. The CAD model was created using SolidWorks 2019 and it is shown in Figure 1a. The computational mesh was created within STARCCM+ using unstructured tetrahedral and polyhedral meshes, which are illustrated in Figure 1b.

To generate the mesh, the grid resolution must meet the criteria based on CFD guidelines. The minimum grid resolution must be set to $\frac{1}{10}$ of the building scale. The expansion ratio should not exceed 1.2 to avoid too high a volume ratio for adjacent cells. The coarser mesh is used for the parts far from the target area. Prism layers are generated close to the walls. The cell size of buildings is 0.5 m, the green internal part in Figure 1 is 1 m and far from the building the grey area is 8 m. There should be a reasonable growth rate between the outer prism layer and the first core cell to avoid too high a volume ratio for adjacent cells. To ensure this, the thickness of the prism layers adjacent to the wall is defined as 10, 20 and 30% of the cell size of each section. The number of prism layers varies between 2 and 5. The number of polyhedral cells created with this size is around 1.4 million. The results of the simulation using the aforementioned settings for grid size were compared with the wind tunnel measurement data.

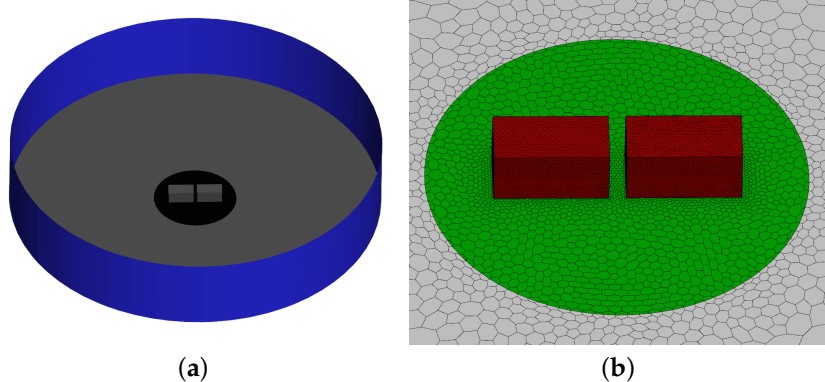

|     |     |
| :-: | :-: |
| (**a**) | (**b**) |

**Figure 1.** (**a**) Computational domain for Test Case 1 used for validation. (**b**) Computational mesh for Test Case 1 used for validation.

### 2.2. Governing Equations and Boundary Conditions

Using RANS methods, the equations that need to be coupled to resolve are continuity and Navier-Stokes. Additional terms are required for closure to calculate the eddy viscosity model, which varies depending on the type of turbulence model. In the case of the k-$\varepsilon$ model, turbulent kinetic energy and turbulent dissipation energy are solved along with the continuity and momentum equations. When using standard k-$\omega$ and k-$\omega$ SST (shear stress transport), specific dissipation rates are defined [12,13]. The SST k-$\omega$ model uses the k-$\omega$ model close to the walls and switches to k-$\varepsilon$ model away from walls [14]. These two are

combined with blending functions. The model was developed due to restrictions of the k-$\varepsilon$ model in the treatment of the near-wall without a damping function. The governing equations of all turbulence models that are used in STARCCM+ v.13.04.010 can be found in [15]. All three models were applied to the geometry consisting of two buildings and a real case of the East Village. The accuracy of all the models was then compared.

In terms of boundary conditions, the computational domain is divided into two sections of inlet and outlet. The boundary condition that is imposed on the flow for the inlet is velocity while for the outlet zero static pressure is imposed [16]. A log-law profile is applied at the inlet and is given in Equation (1). The friction-free velocity ($u^*$) must be calculated based on the reference velocity. Setting the reference wind speed as 5.9 m/s at the pedestrian level of 2 m, $u^*$ is calculated as 0.587 m/s [17]. This velocity is required to define turbulent parameters at the boundaries. It must be noted that in this equation the effect of zero displacement is neglected. For wind speed profiles over rough terrains such as forests, the concept of zero displacement is of importance [18,19]. Since there is no accurate method that can be applied to determine the displacement length, the knowledge of $u^*$ and $z_0$ completely defines the state of the wind. Thus, the effect of vegetation in Section 2.4 is expressed as a drag exerted on the surface, which is similar to the concept of zero-plane displacement [20].

$$\mathbf{u}(z) = \frac{u^*}{k} \ln\left(\frac{z + z_0}{z_0}\right) \tag{1}$$

In the case of using k-$\varepsilon$, turbulent kinetic energy and dissipation energy are defined as follows [21]:

Turbulent kinetic energy:

$$k(z) = \frac{u^{*2}}{\sqrt{C_\mu}} \sqrt{C_1 \ln\left(\frac{z + z_0}{z_0}\right) + C_2} \tag{2}$$

Turbulent dissipation energy:

$$\varepsilon(z) = \frac{u^{*3}}{k(z + z_0)} \sqrt{C_1 \ln\left(\frac{z + z_0}{z_0}\right) + C_2} \tag{3}$$

In the case of using the k-$\omega$ model, specific dissipation rate is given by Equation (4) [14]:

$$\omega(z) = \frac{u^{*3}}{k\sqrt{C_\mu}} \sqrt{C_1 \ln\left(\frac{z + z_0}{z_0}\right) + C_2} \tag{4}$$

In the equations for inlet boundary conditions, $k$ is the von Karman constant, which has the value of 0.41. $C\mu$ is a constant parameter set to 0.09 for the standard k-$\varepsilon$ model. $z_0$ is the aerodynamic roughness length, which has the value of 0.3. $C_1$ and $C_2$ are 1.99 and 1.44 in the case of using the k-$\varepsilon$ turbulence model. $C_t$, which is required for the eddy viscosity model in k-$\varepsilon$, is 1. The boundary condition at the top wall is a free slip wall. For solving the boundary layer close to the walls, the standard wall functions in combination with the sand-grain roughness modification are used [22]. The building walls and roofs are defined as smooth walls, while the ground is described as a rough wall.

The roughness function for the rough surfaces in STARCCM+ is defined by Equation (5).

$$r = \frac{E z_0}{C} \tag{5}$$

where $r$ is the roughness height and is calculated based on the aerodynamic roughness length. These roughness parameters are given to all the walls.

For building walls and roofs, the aerodynamic roughness length is zero and for the ground, it must be defined. The default values of the wall parameters in STARCCM+ are 9 and 0.253 for E and C, respectively [15]. The default values can be changed to reach the

desired roughness height. The roughness height should be $\min(30z_0, 1/2y_p)$ [17], where $y_p$ is the distance between the centre point of the wall-adjacent cell to the wall and $K_s$ is the roughness height. This implies that $y_p$ must be bigger than $2K_s$ ($y_p > 2K_s$).

The pressure–velocity coupling is combined with a segregated flow solver using a SIMPLE-type algorithm. Second-order discretisation schemes are used for all convection terms.

### 2.3. Test Case 2: East Village of London Olympic Park

2.3.1. Computational Domain and Grid

In this section, the wind assessment is carried out on a 3D model representing part of the development in the East Village in the London Olympic Park. The schematic of the CAD model and the location of this development are shown in Figure 2. A 3D model of this geometry was constructed using SolidWorks v.2019 software. The surface of the cylindrical computational domain is divided into 12 equal segments. The height of buildings varies between 17 and 102 m. The diameter and the height of the computational domain are 3 km and 500 m, respectively, which meet the criteria of the CFD guidelines and are large enough to avoid reverse flow pressure. It has been determined from the simulation of two buildings (Test Case 1), that that the results using a polyhedral mesh type are closer to the measurement data compared to a tetrahedral mesh. Thus, for this case, the same mesh type is used. The cell sizes for the buildings, internal subdomain and far from the buildings are 1.5, 3 and 24 m, respectively. The number of cells created is around 6 million. This number of cells was found to be fine enough based on the mesh independency test.

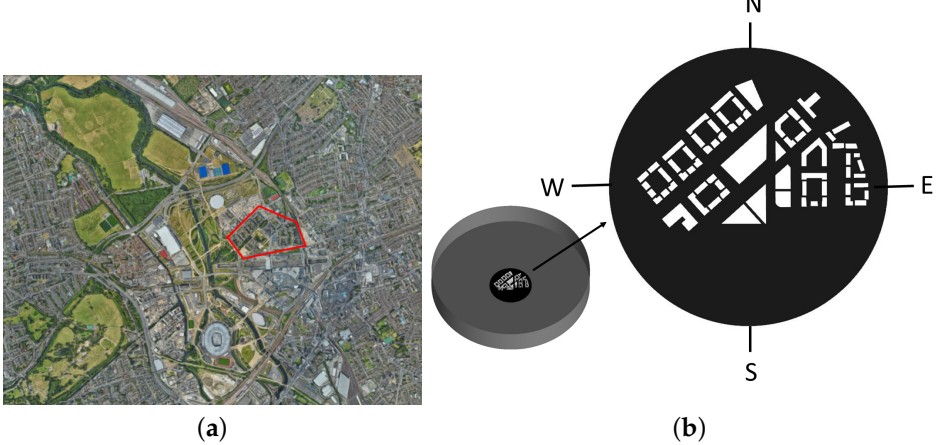

(**a**)　　　　　　　　　　　　　(**b**)

**Figure 2.** (**a**) The East Village location within the London Olympic Park. (**b**) The CAD model generated for the East Village.

2.3.2. Wind Data Analysis and Boundary Conditions

The wind speed is given in different directions at 10 m above the ground level using a wind rose, which is shown in Figure 3b. The cumulative wind speed with different frequency was averaged over the course of 10 years from 2001–2010. This value was used as a reference velocity in the simulation and was calculated as 8 m/s. Based on wind data analysis, the surface of the cylinder in the computational domain is divided into 12 sections, which shows different wind speeds in different directions. In every simulation 6 sections are defined as an inlet and the rest as an outlet. The boundary condition at the inlet of the computational domain is based on the velocity data obtained from the UK Met Office for the closest weather station. It can be observed from the wind rose in Figure 3b that velocity is dominant in the south-west (SW) direction and at an angle of 240°, it reaches the maximum. The inlet boundary is then defined in the range of SW − 90 °C < Inlet < SW + 90 °C. This is shown schematically in Figure 3a. It can be observed that half of the circle is defined as an inlet and the other half is defined as an outlet. The boundary conditions on

the inlet, outlet, top of the computational domain, building walls and roof are the same as the validation case. The data from the weather station is given for the height of 10 m. The aerodynamic roughness length is set to 0.3 for all wind directions, which is an estimated value for suburban or industrial areas [23,24]. Given that the velocity at a height of 10 m is 8 m/s and with aerodynamic roughness length of 0.3, friction-free velocity is calculated as 1.078 m/s.

This value of friction-free velocity is used in turbulent kinetic and dissipation energy for the inlet.

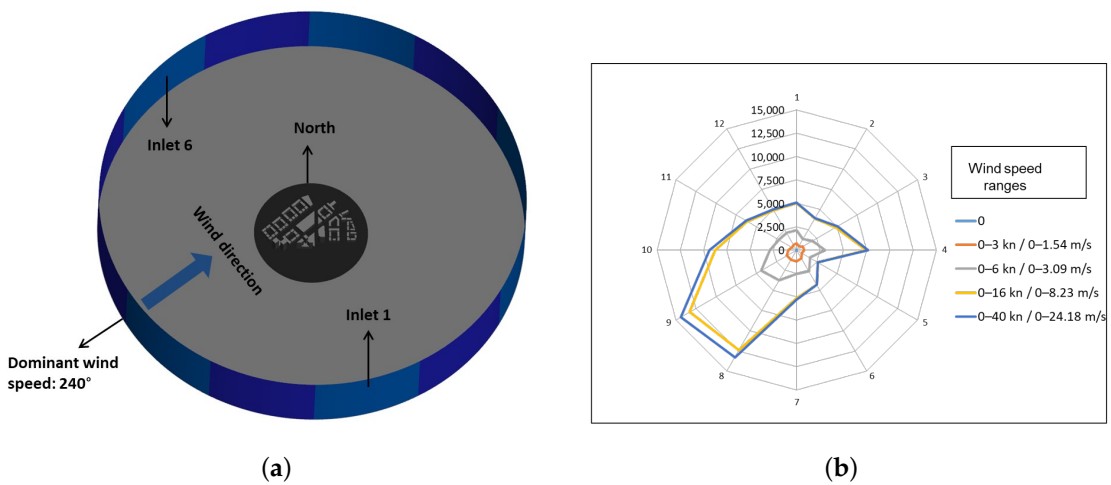

(**a**)                                                                                    (**b**)

**Figure 3.** (**a**) Computational domain for East Village. (**b**) Wind data analysis.

### 2.4. Test Case 3: East Village of London Olympic Park with Vegetation

Planting trees in urban areas contributes to urban heat island mitigation and can reduce energy use, wind speed and air pollution [25]. Urban trees are modelled through implicit and explicit approaches. Using implicit approaches, trees are considered in surface parameterisation with the value of aerodynamic roughness length $Z_0$, which is applied in wall function. However, in the explicit approach, the aerodynamic effects of trees are modelled by adding source and sink terms to momentum, turbulent kinetic energy and turbulent dissipation energy [25]. Extra terms in the equation of turbulence kinetic and dissipation energy account for the enhanced production of turbulence, i.e., wake turbulence due to its smaller scale than shear turbulence, which is subjected to faster dissipation. Thus, vegetation acts as a net sink for turbulent kinetic energy [26,27].

In the present work, trees were added to the case of the East Village with 14 different arrangements. The distance between trees, the proximity to building edges and the type of trees (evergreen or deciduous) will all affect the results. In this study, the trees were modelled using the explicit approach and were considered as porous media. The source and sink terms in the momentum, turbulent kinetic energy and turbulent dissipation energy are activated when the flow reaches the porous media zone according to Equations (6)–(8) (the terms in the boxes indicate the sink and source terms on trees) [26].

$$\frac{\partial u_i}{\partial t} + u_j \frac{\partial u_i}{\partial x_j} = \frac{\partial}{\partial x_j}[\nu(\frac{\partial u_i}{\partial x_i} + \frac{\partial u_i}{\partial x_j}) - \overline{u_i u_j}] - \frac{1}{\rho}\frac{\partial p}{\partial x_i} - \boxed{C_D a|u|u_i} \tag{6}$$

$$\frac{\partial k}{\partial t} + u_j \frac{\partial k}{\partial x_j} = \frac{\partial}{\partial x_j}[(\nu + \frac{\nu_t}{\sigma_k})\frac{\partial k}{\partial x_j}] + P_k - \varepsilon - \boxed{C_D a(\beta_p|u|^3 - \beta_d|u|k)} \tag{7}$$

$$\frac{\partial \varepsilon}{\partial t} + u_j \frac{\partial \varepsilon}{\partial x_j} = \frac{\partial}{\partial x_j}[(\nu + \frac{\nu_t}{\sigma_\varepsilon})\frac{\partial \varepsilon}{\partial x_j}] + P_k - \varepsilon - \boxed{C_D a(\beta_p c_{\varepsilon 4}|u|^3 \frac{\varepsilon}{k} - \beta_p c_{\varepsilon 5}|u|\varepsilon)} \tag{8}$$

where $C_D$ is the drag coefficient and 'a' is the leaf area density, $|u|$ refers to the velocity magnitude and $u_i$ is the velocity component of direction $i$. Given many trees in the UK

are deciduous, the average value for leaf area density is fixed as 1.6. The average leaf area densities for deciduous trees are approximated between 1.06 and 2.18 $m^3 m^{-3}$ [28]. The drag coefficients for most types of vegetation are between 0.1 and 0.3. The constant parameters for tree modelling are defined in Table 1. $\beta_p$ is the fraction of mean kinetic energy that is converted to the wake turbulence, $\beta_d$ is the coefficient that accounts for short-circuiting of eddy cascade and $c_{\varepsilon 4}$ and $c_{\varepsilon 5}$ are empirical coefficients [25]. These parameters can be slightly different for various types of vegetation [29]. The tree stem is not considered in the modelling due to its small size and thickness. The tree stem must be considered as a wall in case of existence, which creates a limitation for meshing. Only the tree crown is set as the porous media and it is elevated from the ground. Depending on the type of tree and its age, tree height, crown width and crown height would be different. To estimate the elevation of porous media from the ground, the tree size must be estimated from empirical correlations. In this study, birch trees were chosen for wind mitigation, which are common trees in urban areas and in the UK [30]. The crown height and width for this tree are estimated by Equations (9) and (10), respectively.

$$Crownheight = \exp(b_0 + b_1 HG + b_2 BHD + b_3 treeage + \beta + \beta_{pt}) \qquad (9)$$

$$Crownwidth = \exp(c_0 + c_1 BHD + b_2 BHD + c_2 treeage + \gamma + \gamma_{pt}) \qquad (10)$$

where *HG* and *BHD* refer to the height growth and the breast height diameter. The coefficients of empirical equations are found from [30]. With the total tree height of 15 m, 2 mm/year height growth at the age of 20 years, the crown height and tree stem width are estimated as 9 m and 6 m, respectively. Applying these coefficients, the crown width is estimated around 4.5 m. Thus, in the CAD model, trees are elevated 6 m above the ground and the tree crown is shown as a rectangular cube with a height of 9 m and a width of 4.5 m in Figure 4. The required constant parameters to find out the tree dimensions are shown in Table 2.

**Table 1.** Constant parameters for tree modelling.

| Constants | $c_d$ | $\beta_p$ | $\beta_d$ | $c_{\varepsilon 4}$ | $c_{\varepsilon 5}$ |
|---|---|---|---|---|---|
| Value | 0.2 | 1 | 4 | 1.5 | 0.4 |

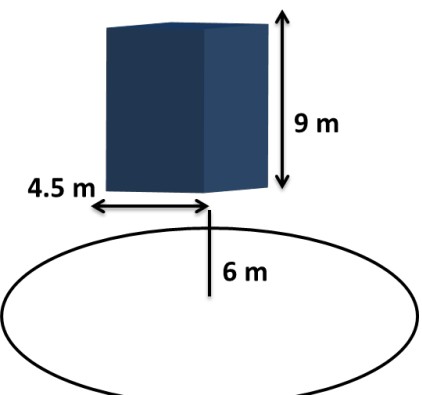

**Figure 4.** Schematic of the trees implemented in the CAD model.

**Table 2.** Constant parameters for estimation of the tree height and crown width.

| Constants | $b_0$ | $b_1$ | $b_2$ | $b_3$ | $c_0$ | $c_1$ |
|---|---|---|---|---|---|---|
| Value | 1.2603 | 0.0468 | −0.0111 | 0.0060 | 0.554 | 0.1596 |
| Constants | $c_2$ | $\beta$ | $\beta_{pt}$ | $\gamma$ | $\gamma_{pt}$ | |
| Value | −0.0141 | 0.02226 | 4.0055 | 0.3156 | 0.8125 | |

## 3. Results and Discussion

### 3.1. Test Case 1

The results of the simulations are presented in terms of velocity profiles at the pedestrian level of 2 m in Figure 5.

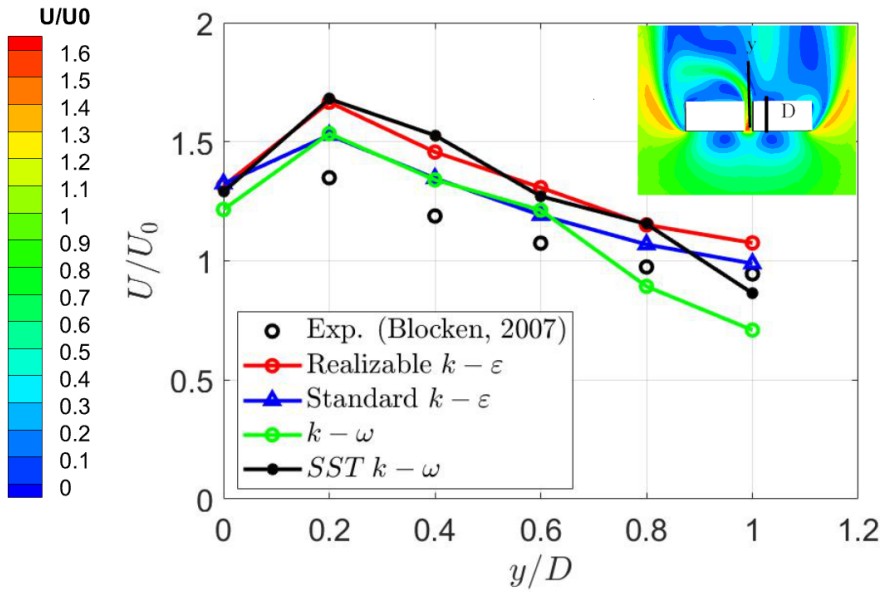

**Figure 5.** Comparison of amplification factors ($u/u_0$): $u_0$: 5.9 (m/s) using different turbulence model (same cell size); building width: 6 m.

The simulation results are for the case with the polyhedral mesh, using a of prism layer thickness of 20% of the core cell size and the number of prism layers as 2. These inputs indicate more accurate results as they fit the measurement data better. Simulation results using different turbulence models are fully converged and the residuals reach below $10^{-4}$. The results show that changing the number of prism layers from 2 to 5 does not affect the simulation. So, to reduce the computational cost for a complex urban geometry it is reasonable to use 2 layers.

For comparing the results of this simulation quantitatively, 5 points within the building passages are taken along the wind direction, as shown in the inset of Figure 5.

It can be seen that overall, the standard k-$\varepsilon$ model fits the experimental results better compared to other models. Nearly all models tend to over-predict the velocity. The standard and SST k-$\omega$ models tend to under-predict the velocity further downstream, therefore predicting a much steeper gradient in the velocity levels with respect to y/D. All the numerical settings used in this rather simple case study provide great insight for the much more complicated case of the East Village, which is presented below.

### 3.2. Test Case 2

Colour contours of velocity at the pedestrian level of 2 m are shown in Figure 6. The area of high velocity is clearly observed from the velocity contours, which are due to corner effects as well as downwash and funnelling effects (this will be discussed further in conjunction with Figure 8).

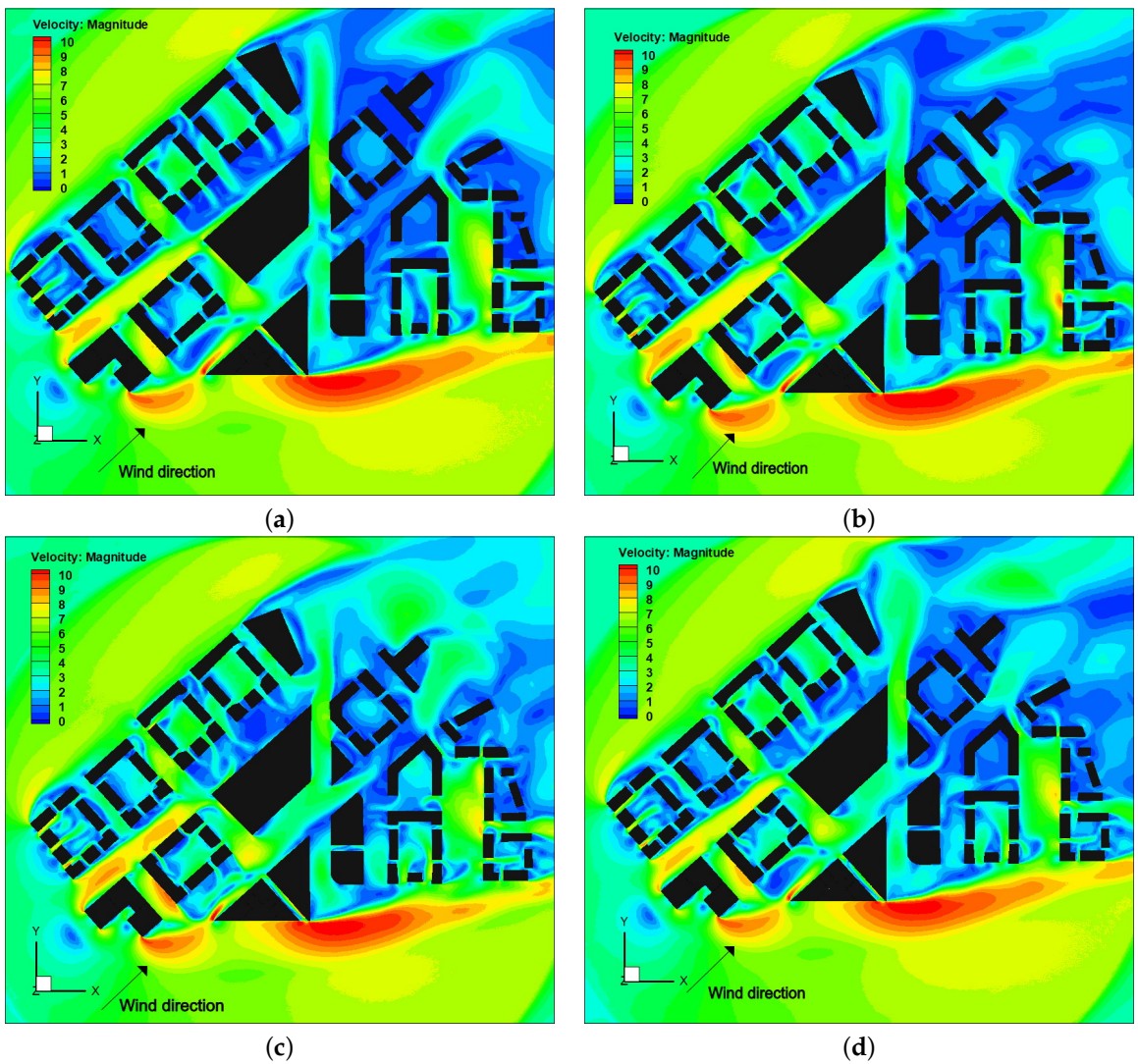

**Figure 6.** Contours of the velocity magnitude: (**a**) standard k-ε, (**b**) realizable k-ε, (**c**) k-ω, (**d**) SST k-ω.

Based on the velocity contours shown in Figure 6, no significant difference between different models can be observed. To further investigate this, the variation of the predicted velocity should be monitored at different heights. To do so, the velocity profiles across three vertical lines are shown in Figure 7 for different turbulence models.

Line 1 is located very close to buildings, just before the wind impinges on the bluff bodies. Along this line, all turbulence models show the most similar trends and the buildings behind them have similar heights. Line 2 is located in the area with the highest velocity magnitudes. The buildings in front of them have heights varying between 30 and 100 m. As can be observed from Figure 7b, velocity variation with height along line 2 and above the height of 30 m are getting similar with a very small variation for all models. Beyond the height of 30 m, the main obstacle is a tall structure and the results appear to exhibit the same trend for four turbulence models. Line 3 is located where the streamlines have passed the bluff bodies. From this line, it is clearly recognised that the k-ω does not obey the logarithmic trend as the velocity decreases up to the height of 30 m, and then it follows an increasing trend. In the work carried out by Tominaga [8], which assessed the accuracy of various turbulence models around one high rise building, it was mentioned that SST k-ω underestimates the turbulent kinetic energy around buildings and as a result, flow separation is expected around the corners of the buildings. Overall, their finding

indicated that k-ε models are more accurate in the prediction of flow around the buildings for both steady and unsteady state RANS simulations.

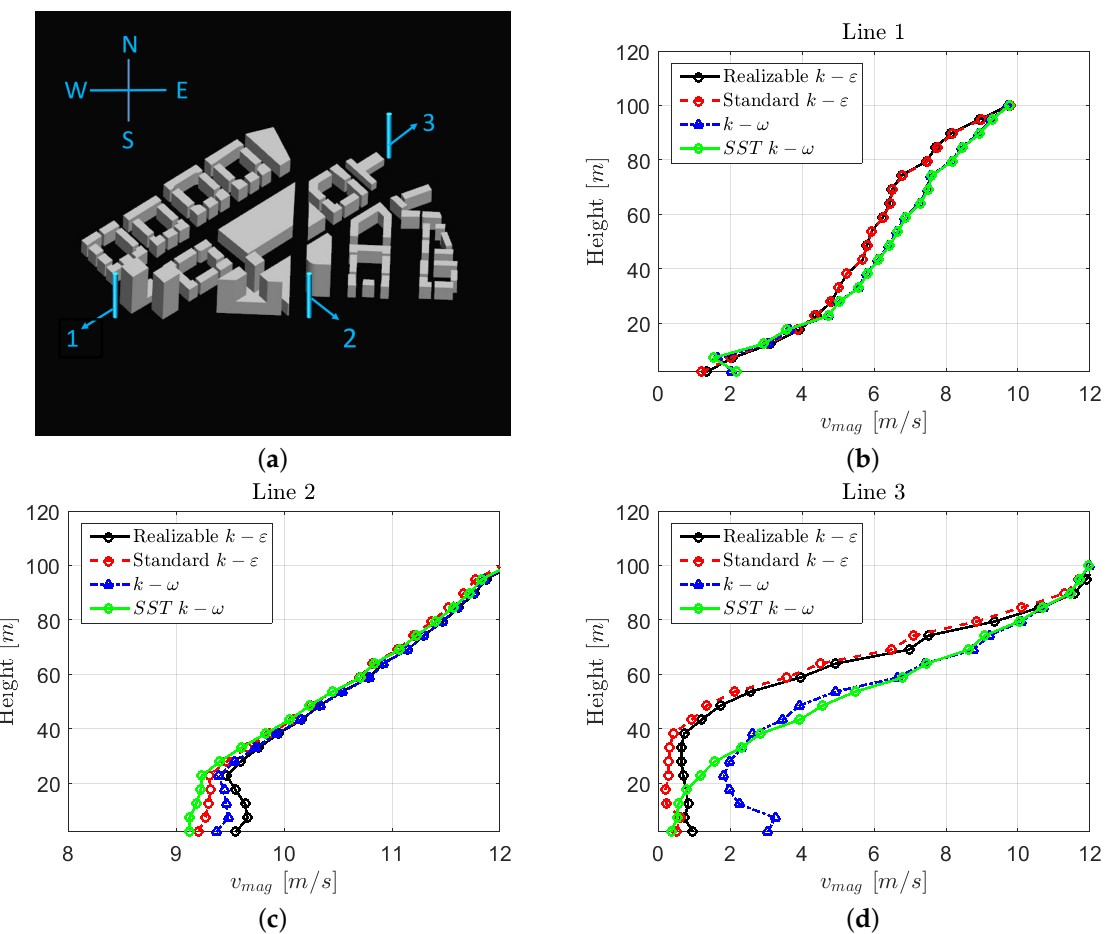

**Figure 7.** Comparison of various turbulent models at different heights for 3 distinct sections of the geometry: (**a**) target lines for comparison of turbulence models, (**b**) line 1, (**c**) line 2, (**d**) line 3.

Despite indicating the velocity contours at pedestrian level, different wind patterns including corner, downwash and funnelling effects are also shown graphically in this section. In Figure 8a, the effect of interaction flow is illustrated. As has been mentioned by Blocken [9], building influence scale is the factor to determine different patterns of flow (e.g., resistance, interaction and isolated flow). This factor is calculated based on Equation (11).

$$S = (B_L B_s)^2 \tag{11}$$

where $S$ is the building influence scale, and $B_L$ and $B_s$ refer to the larger and smaller dimensions of windward faces. In Figure 8a, the distance between buildings is 6 m and the ratio of width to building influence scale (W/S) is in the range of 0.125 < W/S < 1.25, which is in the category of interaction flow. This factor indicates that the streams at the corner entrance separate and subsequently merge into one single stream. In addition, flow separation, corner and downwash effects are illustrated with streamlines in Figure 8a–c, representing areas of high velocity.

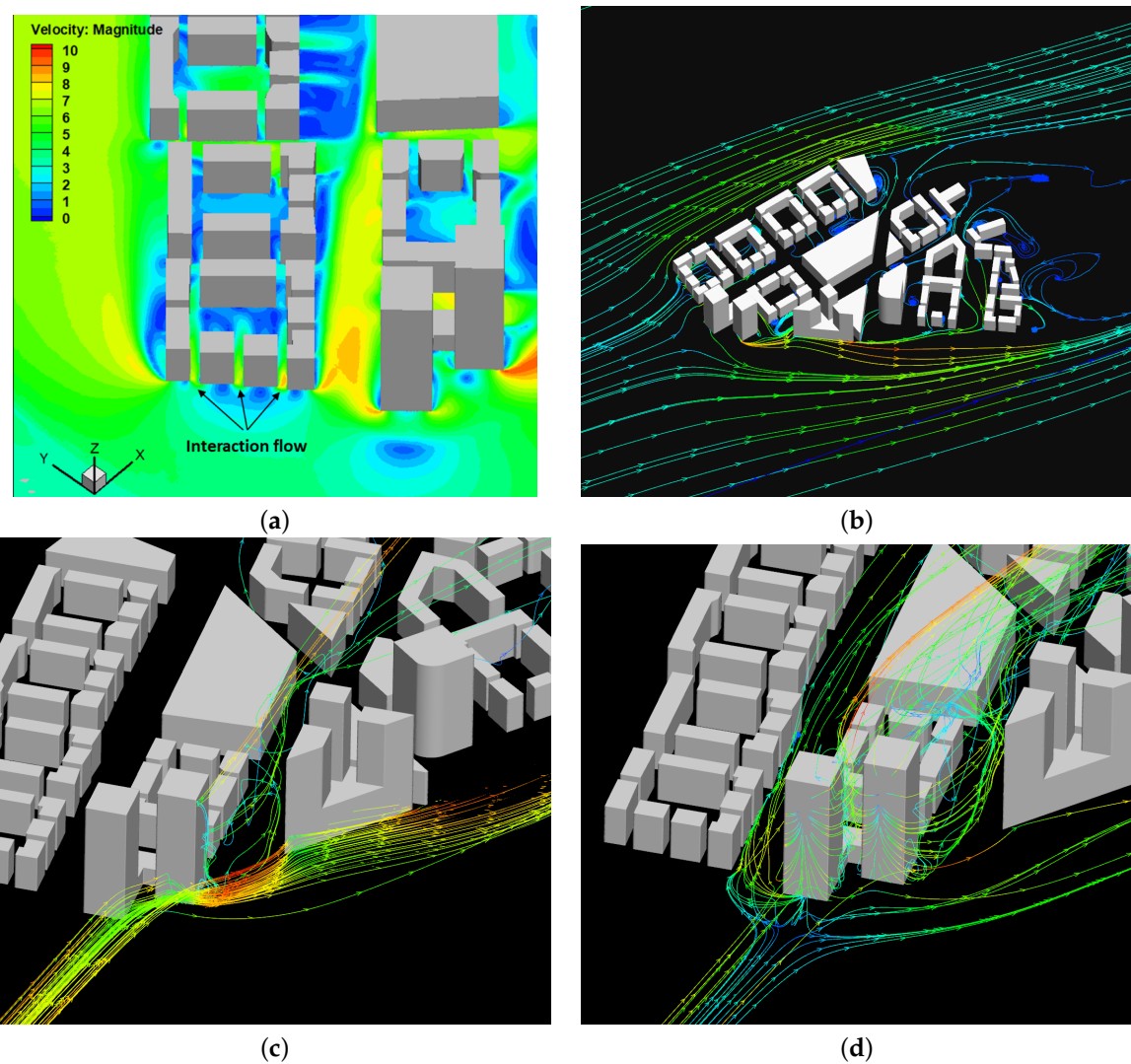

**Figure 8.** Velocity streamlines: (**a**) interaction flow, (**b**) flow separation, (**c**) corner effect, (**d**) downwash effect.

Streamlines in Figure 8b show the flow separation. Where the flow is separated from the surface of the buildings after hitting them it forms a recirculation zone after passing them. Streamlines in Figure 8b demonstrate the corner effect where the wind is accelerating around the corners of the buildings. The high-velocity area near the ground is due to corner effects and this can lead to pedestrian discomfort, while calmer stagnation regions are at the sides of the buildings. In addition, high-velocity areas can occur due to the funnelling effect, where the flow accelerates through narrow passages between buildings. Figure 8c shows the downwash effect. When the wind strikes a tall building, it can flow above or around it and can be partly detached towards the ground, and its intensity depends on the height of buildings. This effect also amplifies the wind speed towards the ground and the pedestrian level.

### 3.3. Test Case 3

The wind comfort level depends on the individual activity [31]. For this reason, Table 3 defines the wind threshold for separate activities. The main objective of using vegetation around buildings is to eliminate A1 regions in the outer boundary of the model for pedestrian comfort, which is presented in Figure 9. In order to remove the A1 regions, which are uncomfortable for pedestrians walking and cycling, the different arrangement of trees

with heights of 12 and 15 m are assessed in this test case. Many factors in tree modelling in urban simulations can affect wind mitigation. Examples include wind direction, tree type (e.g., deciduous, evergreen), tree age, stem height, tree height, crown height, crown width, diameter at breast height, distance between trees and distance of buildings to the trees [25,26,32]. Considering all these parameters simultaneously to find the optimised type of tree and arrangement is beyond the scope of this study. Thus our study deals with optimizing the arrangement of trees after tree selection (e.g., based on price, weather condition). Regions 1–3 in Figure 9 show the main areas in need of mitigation. Trees with a height of 15 m represent a birch type and are elevated 6 m above the ground. Birch trees are deciduous trees and are common in the UK. Only the tree crown, represented as a porous medium, is modelled in this study. The sink and source terms for turbulence and the parameters used for the modelling of vegetation are discussed in Section 2.4. The description of different tree arrangements tested here is shown in Table 4.

**Table 3.** Lawson comfort scale.

| Wind Speed Category | Threshold Wind Velocity [m/s] | Activity |
|---|---|---|
| A4 | 4 | Uncomfortable for pedestrians in the vicinity of entrance door or sitting outside for long period of time. |
| A3 | 6 | Uncomfortable for pedestrians standing or sitting for shorter periods of time. |
| A2 | 8 | Uncomfortable for pedestrians 'leisure walking' e.g., strolling and sightseeing |
| A1 | 10 | Uncomfortable for pedestrians walking quickly e.g., walking to a destination and cycling |

**Table 4.** Description of different tree arrangements tested.

| Case Number | Tree Height (H) [m] | Minimum Distance to Building | Distance to Other Trees | Arrangement |
|---|---|---|---|---|
| Case 1 | 12 | 2/3H | 2H | Individual trees |
| Case 2 | 12 | H/2 | H | Individual trees |
| Case 3 | 12 | H/2 | 2H | Individual trees |
| Case 4 | 12 | H/4 | H | Individual trees |
| Case 5 | 15 | H/2 | 2H | Individual trees |
| Case 6 | 15 | H/2 | H | Individual trees |
| Case 7 | 15 | H/4 | 2H | Individual trees |
| Case 8 | 15 | H/4 | H | Individual trees |
| Case 9 | 15 | H/4 | adjacent | block |
| Case 10 | 15 | H/4 | adjacent | block |
| Case 11 | 15 | H/4 | adjacent | block |
| Case 12 | 15 | H/4 | adjacent | block |
| Case 13 | 15 | H/4 | H/4 | block |
| Case 14 | 15 | H/4 | H/4 | block |

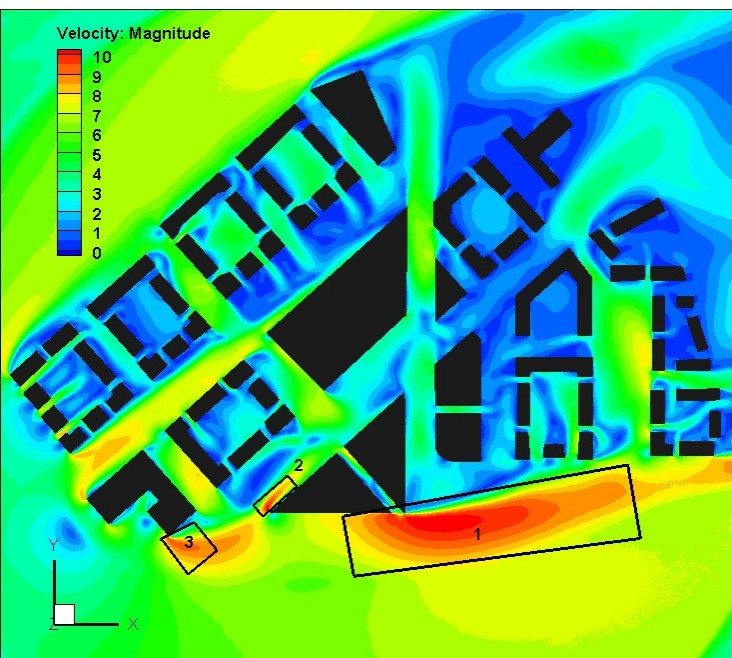

**Figure 9.** A1 region inside the boundary: targeted area for wind speed mitigation.

The velocity contours at the pedestrian level for the 12 m tree for different tree arrangements are shown in Figure 10 (cases 1–4). The crowns of these trees are modelled as a cube with 5 m in width and 8 m in height and are elevated 4 m above the ground. The tree crown is closer to the ground in comparison to birch trees (Cases 5–8). By looking at Cases 1 to 4, it is evident that the A1 region is smaller when the distance between the trees is reduced. However, the minimum distance between trees and buildings is not large enough to overcome the corner acceleration that leads to the A1 region. The minimum gaps between trees and buildings for trees with a 12 m height are set as 3, 6 and 8 m. Changing the type of trees with bigger crown width and higher leaf area intensity can lead to improved performance.

The velocity contours for birch trees are shown in Cases 5–8. By comparing Cases 5 and 6, no significant difference can be observed, despite the trees in Case 6 being denser. The results suggest that the porous media that is closer to the ground works more effectively by comparing Cases 2 and 6 for tree heights of 12 and 15 m. Both configurations have the same density for trees and the same minimum distance to buildings. Even though the crown size is bigger for the 15 m tree, due to the proximity of porous media to the ground, 12 m trees are more effective at decreasing the wind velocity. In general, 15 m trees work less effectively than 12 m trees for all the cases assessed here because the tree crown is located higher above the ground. In Figure 11, various blocks have been defined as a representation of adjacent trees, an approach which is very efficient in terms of computational time [26,33]. By comparing 14 cases of wind mitigation in this study, it appears that tree arrangements from Cases 9 to 14 act more effectively in terms of wind velocity mitigation at the pedestrian level. However, to compare the effectiveness of Cases 9 to 14, the area-weighted averages of velocity are taken for regions 1–3 and are presented in Table 5.

The results in Table 5 demonstrate that the most effective arrangement for wind speed reduction at the pedestrian level for region 1 is Case 14, leading to approx. 25% reduction. For regions 2 and 3, Case 12 leads to 66 and 3% reductions, respectively. As can be observed from the results, in region 3, despite inserting blocks all around the buildings, the wind velocity cannot be decreased significantly. This suggests that in this region, other types of trees with perhaps wider crowns should be considered.

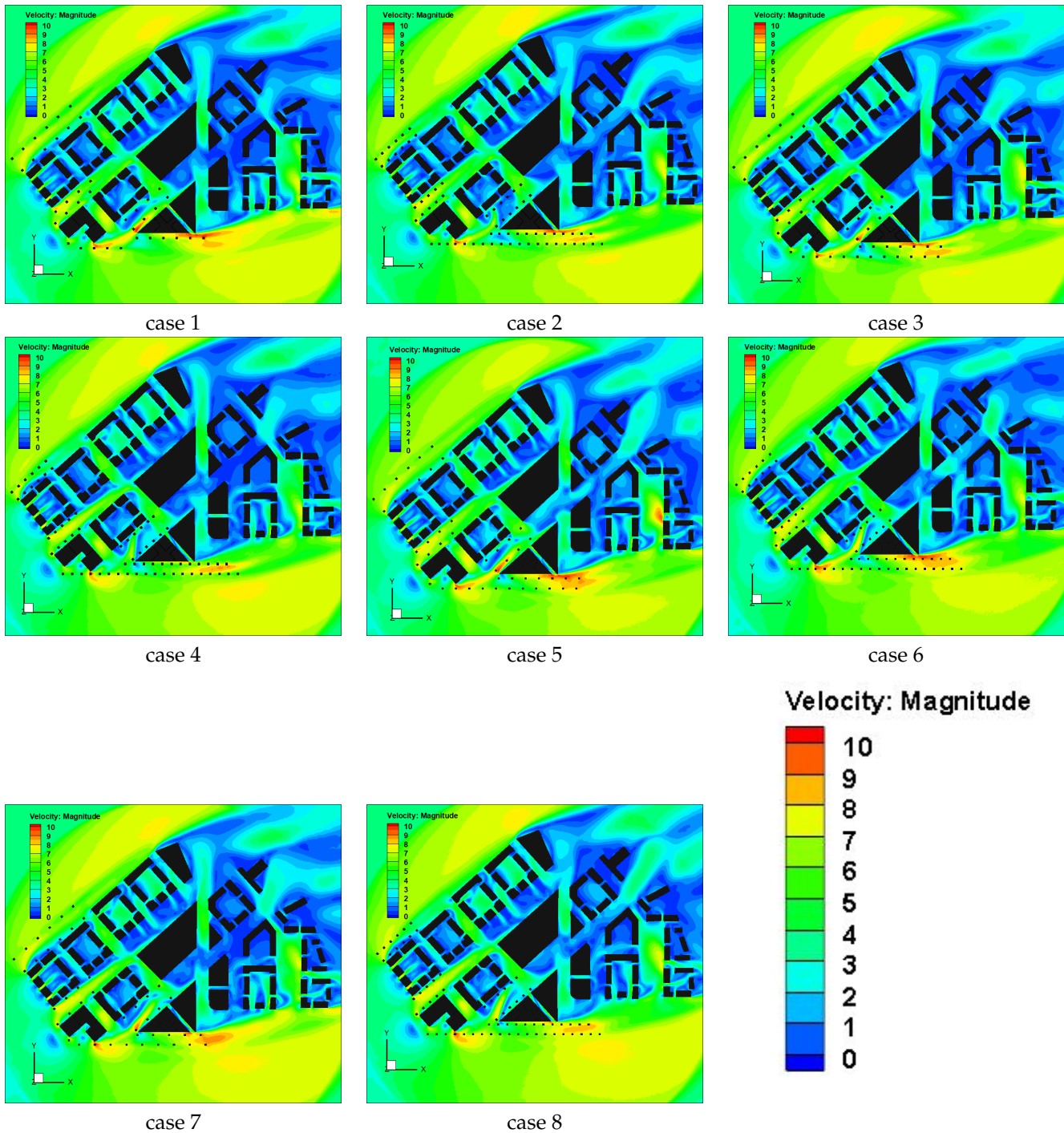

**Figure 10.** Velocity contours at the pedestrian level (2 m) with various tree arrangements: (**case 1**) tree height: 12 m, minimum distance to buildings: 2/3H, distance to other trees: 2H, (**case 2**) tree height: 12 m, minimum distance to buildings: H/2, distance to other trees: H, (**case 3**) tree height: 12 m, minimum distance to buildings: H/2, distance to other trees: 2H, (**case 4**) tree height: 12 m, minimum distance to buildings: H/4, distance to other trees: H, (**case 5**) tree height: 15 m, minimum distance to buildings: H/2, distance to other trees: 2H, (**case 6**) tree height: 15 m, minimum distance to buildings: H/2, distance to other trees: H, (**case 7**) tree height: 15 m, minimum distance to buildings: H/4, distance to other trees: 2H, (**case 8**) tree height: 15 m, minimum distance to buildings: H/4, distance to other trees: H.

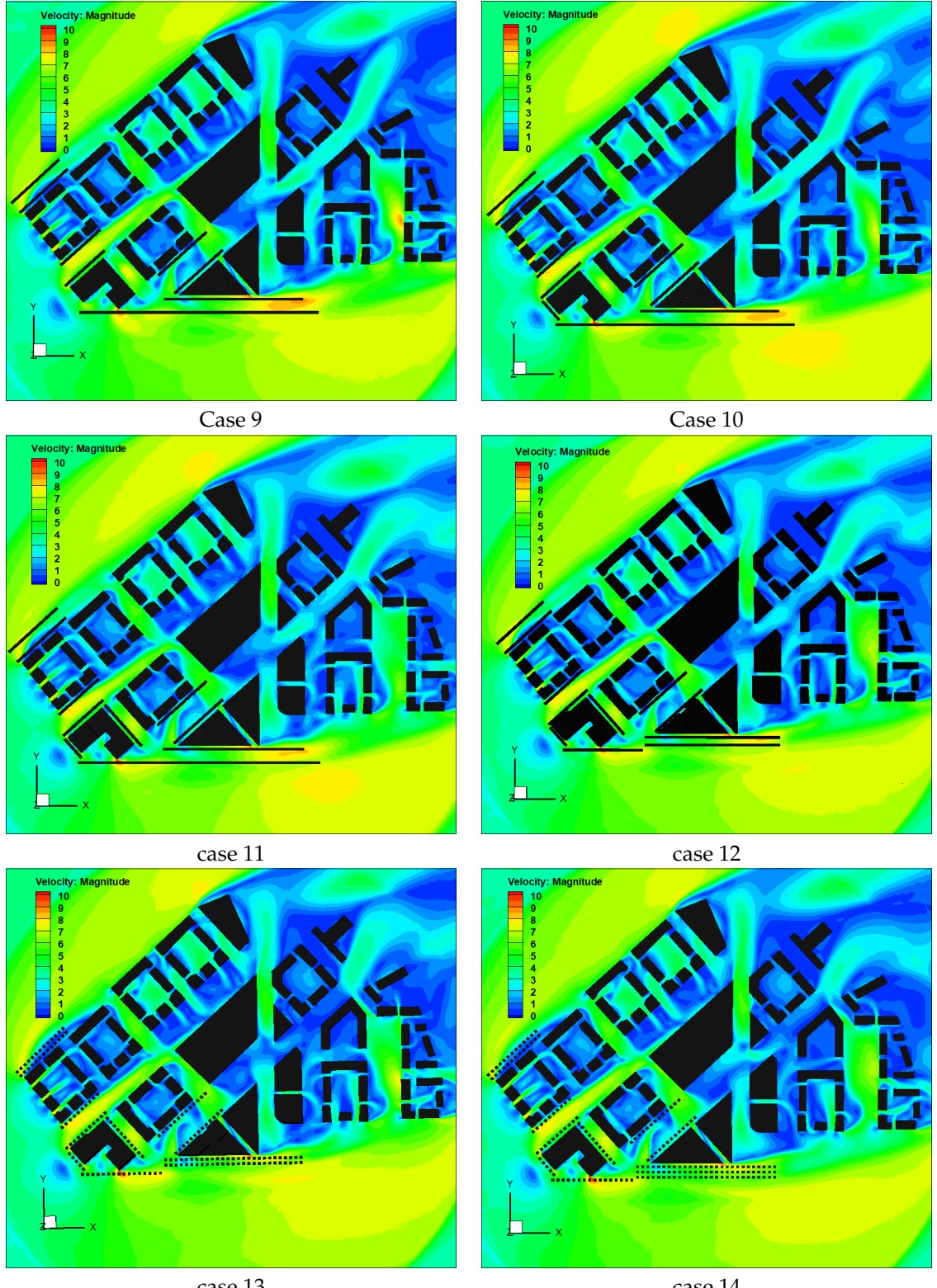

**Figure 11.** Velocity contours at the pedestrian level (2 m) with various tree arrangements and tree height of 15 m: (**case 9**) minimum distance to buildings: H/4, distance to other trees: adjacent, (**case 10**) minimum distance to buildings: H/4, distance to other trees: adjacent, (**case 11**) minimum distance to buildings: H/4, distance to other trees: adjacent, (**case 12**) minimum distance to buildings: adjacent, distance to other trees: 2H, (**case 13**) minimum distance to buildings: H/4, distance to other trees: H/4, (**case 14**) minimum distance to buildings: H/4, distance to other trees: H/4.

**Table 5.** Area weighted average of velocity for regions 1, 2 and 3.

| Case Number | Area Weighted Average of Velocity (m/s) | | |
|---|---|---|---|
| | Region 1 | Region 2 | Region 3 |
| East Village without trees | 8.05 | 8.80 | 5.40 |
| Case 9 | 7.06 | 2.83 | 5.28 |
| Case 10 | 7.04 | 2.88 | 5.33 |
| Case 11 | 6.84 | 2.84 | 5.51 |
| Case 12 | 6.71 | 2.68 | 5.24 |
| Case 13 | 6.68 | 2.93 | 5.64 |
| Case 14 | 6.03 | 3.12 | 5.61 |

It is worth noting that a standard wall function was used in Test Cases 2 and 3 in contrast to the work of Blocken [9], where the wall function roughness was modified. However, further work is required to eliminate the effect of horizontal inhomogeneity such as using periodic boundary conditions instead of Richard and Hoxey [21], including boundary conditions for the velocity profile, turbulent kinetic and dissipation energy. Structured mesh could also improve the efficiency of CFD simulations. However, to reduce the computational cost, a structured mesh in the area of interest and unstructured far from buildings is more desirable. Besides, unsteady-state RANS simulation for this case may lead to improved prediction of the wind speed [8]. For reducing uncertainty, higher fidelity CFD techniques can be used (e.g., LES versus RANS) [34,35]. However, using an expensive method of LES might not be very feasible for complex urban geometries such as the ones tested here due to significant computational cost [36]. Therefore, RANS still appears to be a preferred choice and a compromise between accuracy and cost in simulating complicated and/or large urban models [37].

## 4. Conclusions

In the present study, CFD simulations were conducted for three different test cases. The first case was a simple model to conduct sensitivity analysis and validating the CFD methodology. Subsequently, the second case involved a large and detailed 3D model representing the East Village in the London Olympic Park. Finally, the third case focused on assessing the effects of mitigating against the wind by planting trees. Many factors can affect wind mitigation including wind direction, tree type (e.g., deciduous, evergreen), tree age, stem height, tree height, crown height, crown width, diameter at breast height, distance between trees and distance of buildings to the trees [25,26]. Considering all these parameters simultaneously to find the optimised type of tree and arrangement is in the scope of a parametric study. Thus, this section of our study dealt with optimising the arrangement of trees after the tree selection (e.g., based on price, weather condition, etc.). The simulations were performed using four turbulence models including standard k-$\varepsilon$, realizable k-$\varepsilon$, standard k-$\omega$ and SST k-$\omega$, building on the previous track-record of the authors in the field [38–40]. By making a comparison between simulation results and measurement data in Test Case 1 and other references, including the work of Tominaga [8], it is believed that the simulation of wind on an urban scale works more effectively with the derivatives of the k-$\varepsilon$ model. However, other renormalized groups of k-$\varepsilon$ might show more accurate results, but in this work, only the available turbulence models in STARCCM+ were tested, which can be used with wall function methods. The following main conclusions can be drawn from this study.

- Unstructured polyhedral mesh gives more accurate results compared to a tetrahedral mesh and increasing the number of prism layers from 2 to 5 does not change the results significantly.
- By using a wall function to predict the velocity around buildings, there should be a reasonable growth rate between the outer prism layer and the first core cell. The

results of the simulation of Test Case 1 show that more accurate results are obtained if the prism layer total thickness is 20% of the core cell size.

- The validation study revealed that the standard and realizable k-$\varepsilon$ turbulence models show more accurate results, while the results of the standard k-$\varepsilon$ were slightly closer to the measurement data.
- The commonly used SST k-$\omega$ model underestimates the turbulent kinetic energy around buildings and as a result, flow separation is expected around the and therefore was found to be less accurate compared to standard and realizable k-$\varepsilon$ models for this application.
- With the optimised arrangement of trees in Test Case 3 using a specific type of trees (e.g., birch), the wind speed at the pedestrian level is reduced by 25% in region 1, 66% in region 2 and 3.5% in region 3.
- The results of Test Case 3 demonstrate that in the case of using birch trees, denser trees are required to overcome the high-velocity areas due to the corner effect. However, if the tree crown is closer to the ground, lesser trees can be planted in those regions. This conclusion demonstrates the effect of tree age. Younger trees with crowns closer to the ground mitigate wind more. However, older trees with wider crowns are able to decrease wind more. More investigation is required to assess the impact of tree age.
- In certain regions with high-velocity wind, using trees with a wider crown, or locating trees closer to the edge of buildings are likely to overcome the corner and downwash effects more efficiently. Further work is required to assess the impact of evergreen trees with wider crown.

**Author Contributions:** Conceptualization, A.H. and A.K., Software, A.H., Methodology, A.H., Validation, A.H., Investigation, A.H., Visualization, A.H., Writing—original draft, A.H., Resources, A.K., Writing—review & editing, A.K. All authors have read and agreed to the published version of the manuscript.

**Funding:** This research received no external funding.

**Institutional Review Board Statement:** Not applicable.

**Informed Consent Statement:** Not applicable.

**Data Availability Statement:** Data sharing is not applicable to this article.

**Acknowledgments:** The first author would like to thank the Department of MACE at the University of Manchester for providing PhD funding under the "Exceptional Women in Engineering" scheme. The authors would also like to acknowledge the valuable contributions of Nima Shokri from the University of Manchester throughout this project.

**Conflicts of Interest:** The authors declare no conflict of interest.

## Nomenclature

| | |
|---|---|
| $a$ | Leaf area density, $\text{m}^2\text{m}^{-3}$ |
| $b_0$ | Input parameter for tree model |
| $b_1$ | Input parameter for tree model |
| $b_2$ | Input parameter for tree model |
| $b_3$ | Input parameter for tree model |
| $c_0$ | Input parameter for tree model |
| $c_1$ | Input parameter for tree model |
| $c_2$ | Input parameter for tree model |
| $c_{\varepsilon 4}$ | Input parameter for turbulent dissipation energy source term |
| $c_{\varepsilon 5}$ | Input parameter for turbulent dissipation energy source term |
| $C$ | Roughness height, m |
| $C_1$ | Constant parameter for k-$\varepsilon$ model |
| $C_2$ | Constant parameter for k-$\varepsilon$ model |
| $C_D$ | Drag coefficient |
| $C\mu$ | Constant parameter for k-$\varepsilon$ model |
| $E$ | Constant parameter in wall function for rough surfaces |

| | |
|---|---|
| $k$ | von Karman constant |
| $K$ | Turbulent kinetic energy, $\text{m}^2\text{s}^{-2}$ |
| r | Constant parameter in wall function for rough surfaces |
| $|u|$ | Velocity magnitude, $\text{ms}^{-1}$ |
| $u_0$ | Reference velocity at height 2 and 10 m, $\text{ms}^{-1}$ |
| $u^*$ | Friction free velocity $\text{ms}^{-1}$ |
| $z_0$ | Aerodynamic roughness length, m |
| Greek Symbols | |
| $\beta$ | Input parameter for tree model |
| $\beta_d$ | Input parameter for turbulent kinetic energy source term |
| $\beta_p$ | Input parameter for turbulent kinetic energy source term |
| $\beta_{pt}$ | Input parameter for tree model |
| $\gamma$ | Input parameter for tree model |
| $\gamma_{pt}$ | Input parameter for tree model |
| $\varepsilon$ | Turbulent dissipation energy, $\text{m}^2\text{s}^{-3}$ |
| $\omega$ | Turbulent dissipation rate, $\text{m}^2\text{s}^{-3}$ |
| Subscripts | |
| $i, j$ | Scalar node position |
| Acronyms | |
| BHD | Breast height diameter, m |
| CFD | Computational fluid dynamics |
| HG | Height growth |
| LES | Large eddy simulation |
| RNG | Renormalisation group |
| RANS | Reynolds-averaged Navier–Stokes |
| *Re* | Reynolds |
| SST | Shear stress transport |
| SW | South west |

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
