# Peer review of "Computational Simulation of Wind Microclimate in Complex Urban Models and Mitigation Using Trees"

_buildings, doi:10.3390/buildings11030112_

Round 1

Reviewer 1 Report

Dear Authors

I revised the manuscript “Computational Simulation of Wind Microclimate in Complex Urban Models and Mitigation using Trees” submitted to the “Buildings” Journal. The paper is very interesting. However, I have some concerns, which need to be addressed.

Title of the article

“Computational Simulation of Wind Microclimate in Complex Urban Models and Mitigation using Trees”

The title of the article correctly characterises the scope of the research and the potential effects of the research.

Abstract

The basic assumptions of the aim and scope of the study are presented. The methodology is signalled in a narrative way. The result of the study is presented in too general way. Presented conclusions from the research are statements which should be supported by a verification parameter. Quality parameters of the models that support the final thesis in the abstract and indicate the methodological basis of the conclusion have not been quoted.

Improving the content of the abstract in terms of the research conclusions will help to increase the interest of the readers.

“……Keywords: Wind Microclimate, Wind assessment, Computational Fluid dynamics, CFD, building engineering, turbulence modelling, urban design…….”

CFD - the short form repeats the wording in 'keywords'. Duplication of keywords is not correct in the presented form.

In my opinion, perhaps a key word was not included for one of the key parameters, which was the use of trees.

Nomenclature Page 3 and Page 4

"....friction free velocity..."

The convention of capital letters in the description of designations and symbols has not been followed.

A column with possible units of measurement of the presented parameters has not been introduced.

"......Input parameter for tree model..." Many of the parameter descriptions are very general which makes identification by the reader difficult.

  1. Introduction: Page 4, Page 5, Page 6

The difference in approach to the research question between previous academic publications and the undertaken research effort is indicated. The genesis of the research topic is clearly indicated.

The aim of the work is indicated correctly and provided with an appropriate commentary to the scope of the work.

  1. Problem Definition and Methods

2.1. Test Case 1: CFD simulation of wind speed between parallel buildings

“…..In the present work, a cylindrical domain is used as com- putational domain in contrast to the work of Blocken where a block was used….”

It is helpful to explain and justify the adoption of an appropriate computational domain in terms of methodology. This will contribute to a better perception of the work by the readers.

2.1.1. Computational domain and mesh

Page 7. The large number of limits and assumptions given for the scope of the study requires additional justification by reference to the state of knowledge each time new methodological guidelines are provided. Please take this into account.

2.2. Governing equations and boundary conditions

Page 8 The limits of the models and the applied boundary conditions are presented in an understandable form.

In spite of the references to formulas and equations (Page 9), the relevant numbers of model equations are not indicated in the text of the chapter. The inclusion of references to mathematical formulae enhances the reader's understanding of the text. Please take this into account.

Page 9. “…..In the equations for inlet boundary conditions….” The reference to mathematical equations in the text requires a precise indication of their order numbers.  Limitations and optimal indicators mentioned in the text require each time the identification of the state of knowledge (normative acts or model experiments) adequate to the presented values. Please take this into account.

2.3. Test Case 2: east village of London Olympic park

2.3.1. computational domain and grid

2.3.2. Wind data analysis and boundary conditions

Page 10, Page 11. The justifications for "cell size" and " the number of used cells" need to be better confirmed by the state of knowledge in the area of methodology. The experiences of other authors in scientific publications supporting the confirmation of the methodological assumptions can always be identified.

“…..The boundary conditions on the inlet, outlet, top of computational domain, on building walls and roof are the same as validation case….” References in the text to any activities, including validation results, should include immediate references to the state of knowledge. This promotes a better understanding of the article by the reader.

2.4. Test Case 3 : east village of London Olympic park with vegetation

“…..In the present work, trees are added to the case of the east village with 14 different arrangements….”

Every time you formulate the scope of your work, please make sure to indicate the reasons for your actions even if you have to duplicate statements. In the case of extensive methodological notes, this helps the reader to maintain his interest in the text.

  1. Results and Discussion

3.1. Test case 1

Figure 5: The figure is unclear. Please consider increasing the format of the figure or presenting it in an alternative form.

I do not notice in subsection 3.1 a typical discussion of the results with references to the state of knowledge.

3.2. Test case 2

Many of the mentioned research effects are not discussed in terms of competitive research. There is also a lack of claims about the innovation of own observations or their secondary characterisation and emulation.

3.3. Test case 3

There is an apparent tendency to limit the discussion of results by presenting a limited number of competitive studies across the spectrum of the work. The final conclusions prepare the ground well for the conclusions of the study.

  1. Conclusion

Conclusions are presented correctly but without references to the observations of competitive research. Conclusions have the character of a summarised description of research results, which slightly reduces their essential (scientific) value.

Reviewer 3 Report

It is a nice piece of work that will be of good interest to wind microclimate community.
Authors simulated flowfield around buildings and investigated mitigation method to reduce wind speed at pedestrian level using trees. I recommend this paper to be accepted after the following minor concerns are addressed.

1)
There are some minor typographical errors, e.g.,
- in page 5, “Yoshihide et al.[8]” --> “Tominaga et al.[8]” (the last name of author(ref8) seems to be Tominaga).
- In page 8, “The accuracy all models are …” --> “The accuracy of all models are …”
- etc.
Please proofread the manuscript to correct them.

2)
In table 3, wind speed category is assigned to C_i (i=1,2..). Same symbol C_i is used for constant parameter for k-epsilon model, which may be confusing for readers. So, it would be better to use any other symbols.

3)
In fig 4, the tree model used in this study has 4.5m x 9m cross section in x and y direction, and half the area (4.5x4.5) in z direction. Here, it seems the amount of wind deceleration by trees in z-direction is less than that in x or y direction. In light of this, it looks reasonable that the velocity magnitude in region 3 in Figs 10 & 11 remained high where wind downwash (mainly in z-direction) is strong as shown in Fig 8d. So, it would be important to show z-velocity component plot along with velocity magnitude in Fig 9 to help understand the flow structure at pedestrian level.

Reviewer 4 Report

In the paper “Computational simulation of Wind Microclimate in Complex Urban Models and Mitigation using Trees”, the authors present a set of Reynolds Averaged Navier Stokes (RANS) simulations of the wind flow around buildings in urban environment. They test a baseline urban scenario and assess the use of trees as shielding elements to decrease the wind speed and avoid potential pedestrian discomfort.

The topic is of interest, but the methodology, presentation of the results and discussion should be improved. I list below my concerns:

  1. The novelty of the paper is not very clear. The use of trees as shielding element is not new in itself, likewise the simulation methodology. The authors do not provide information or insight on the selection of the tree placement either, and one is left wondering what is the main message of the paper.
  2. Converge of the simulations, Figure 5. The authors mentioned that the residuals are lower than a numerical threshold, however this is not sufficient to guarantee that the simulation results are converged. The inset in figure 5 shows that the wake downstream buildings is not symmetric, which I suspect it is due to lack of convergence. Have the authors performed a convergence analysis? Have the authors considered whether an unsteady computation is more appropriate for these simulations?
  3. Figure 5, comparison against the experimental results. The error is quite large for all the turbulence models, and the setup can hardly be said to be validated. Have the authors assessed whether their mesh is sufficiently fine for this case? I understand that the authors have used guidelines in the literature to design the mesh, but a mesh convergence study should be performed nonetheless.
  1. Page 17, first line of section 3.2. I cannot find any mesh independency test results in figure 6 as the authors indicate in the text. Rather figure 6 compares the different turbulent models and present significant differences. The authors surmise in the text that the k-omega SST has the poorest performance; however, it is hard to determine from figure 6 whether a model is outperforming another because there is a significant scatter. Has a mesh convergence analysis been performed? Which model have the authors selected as the most reliable for the study in figure 6 and why?

  2. Table 5 and figures 10-11. How was the tree placement determined? Is there any guideline which could be of use for a general case?

  3. The authors considered only one wind direction as the wind distribution at the site is relatively narrow and peaks at one single wind direction. However, have the authors considered any potential downside in pedestrian comfort due to the presence of trees for other wind directions?

  4. In their conclusions list, the authors provide a number of computational guidelines, which however do not seem very generalizable. At bullet point 3, the authors state that the k-epsilon models provide more accurate results, but key-omega SST is close to the experimental data. On what basis then the k-epsilon models have “more accurate” results, if this is not based on the measurement? Also there is not support in the paper for the next bullet point (k-omega underestimating TKE – I could not see any plot of turbulent kinetic energy).
  5. There are several typos throughout the manuscript, including misspelled references, misplaced figure labels, which at times make the discussion hard to follow at all. Extensive editing is required.

Round 2

Reviewer 2 Report

I do NOT think the authors give full consideration to my previous comment major concern #1, and the reply is not convincing. I insist that, theoretically, Equation 1 is not fully correct. The correct expression should involve the zero-plane displacement and roughness length. Only the roughness length (z_0)is NOT enough to represent the turbulent flow interactions with the roughness. The effects from the zero-plane displacement can make very huge differences to the results. If the authors insist on this point, please add a detailed formulation derivation in the appendix to show how Equation 1 can be derived from the original turbulent equations.

Reviewer 4 Report

I thank the authors for addressing my concerns.
I believe the answers are satisfactory and the grid-independence studies are helpful to assess the simulations results.

I have a few minor comments

  1. Page 17, first line of paragraph 3.2: "The results of the mesh independency test are demonstrated at the pedestrian level of 2m in figure 6." I believe the wording is misleading here, because figure 6 does not show the mesh-independency test results (which were shown in the rebuttals). Perhaps something along the line of "Color contours of velocity at the pedestrian level of 2m are shown in fig.6" would more descriptive of the figure
  2. Page 30, "Conclusion" section: "Thus, this section of our study dealt with...". Perhaps "Thus, our study dealt with..." is more appropriate in the conclusions section of the manuscript

Round 3

Reviewer 2 Report

I appreciate the detailed feedback from the authors. And the authors gave full considerations to the comments and provided more detailed explanations.  For the formulation of the log-law profile (Equation 1), the authors may keep the current form, it is suggested to mention the existence of zero-plane displacement $d$, refer to some paper using log-law profile with $d$,  and short comments about why your formulation does not have this term (similar as the feedbacks to the reviewer). In summary, the paper could be suggested for publication with minor revisions.

Author Response

"Thanks for your comment. We have now added a paragraph in the manuscript on page 9, briefly addressing the point raised on the zero-plane displacement. We hope this is satisfactory and once again we appreciate your valuable feedback."